# Royal Jelly Protected against Dextran-Sulfate-Sodium-Induced Colitis by Improving the Colonic Mucosal Barrier and Gut Microbiota

**DOI:** 10.3390/nu14102069

**Published:** 2022-05-15

**Authors:** Jianying Guo, Baochen Ma, Zixu Wang, Yaoxing Chen, Wenli Tian, Yulan Dong

**Affiliations:** 1Key Laboratory of Precision Nutrition and Food Quality, Ministry of Education, College of Veterinary Medicine, China Agricultural University, Beijing 100193, China; gjy2019@cau.edu.cn (J.G.); zxwang@cau.edu.cn (Z.W.); yxchen@cau.edu.cn (Y.C.); 2China Animal Husbandry Group, Beijing 100070, China; mabc@cahg.com.cn; 3Institute of Apicultural Research, Chinese Academy of Agricultural Sciences, Beijing 100093, China

**Keywords:** royal jelly, inflammatory bowel disease, colitis, intestinal mucosal barrier, gut microbiota

## Abstract

Royal jelly (RJ) is a natural bee product that contains a variety of biologically active ingredients and has antitumor, antiallergic, antibacterial and immune-regulating effects. Inflammatory bowel disease (IBD) is a chronic inflammatory disease of the intestine that can cause abdominal pain and diarrhea. With this study, we aimed to explore the protective effect of RJ on DSS-induced colitis in mice. The physiochemical parameters (water, protein, 10-hydroxy-2-decenoic acid, total sugar, starch, ash and acidity) of the RJ samples used in this study met the requirements of the international and Chinese national standards. Treatment with RJ improved symptoms and colonic cell apoptosis and decreased intestinal permeability by increasing the expression of tight-junction protein, goblet cells and their secretion mucin, MUC2, in DSS-induced ulcerative colitis mice. RJ also reduced the expression of proinflammatory cytokine IL-6 and increased the expression of anti-inflammatory cytokine IL-10 and sIgA. DSS resulted in an increase in the relative abundance of *Parabacteroides*, *Erysipelotrichaceae*, *Proteobacteria* (*Gammaproteobacteria*, *Enterobacteriales* and *Enterobacteriaceae*) and *Escherichia Shigella* in the colon and a decrease in the relative abundance of *Muribaculum*. In the RJ treatment group, the relative abundance of the above intestinal flora was improved by treatment with 2.0 g/kg RJ. These results suggested that RJ alleviated DSS-induced colitis by improving the colonic mucosal barrier.

## 1. Introduction

An increasing number of natural products are used to treat health problems at present, many of which have been proven to reduce the risk levels of patients with chronic diseases, as well as necessary investments in health care [1,2]. In the past few years, natural bee products have been widely applied in traditional and modern medicine. Bee products such as bee pollen (BP), propolis, bee bread (BB), royal jelly (RJ) and beeswax (BW) can all function as active pharmaceutical ingredients. Royal jelly is marketed on a large scale in Asia, especially in China and Japan, as a health food and cosmetic product [3,4]. As a bee product, royal jelly (RJ) is used to maintain human health [5]. Through continuous efforts, China has cultivated high RJ-producing bees (RJBs) from Italian bees. The royal jelly production of RJBs accounts for more than 90% of the world’s total production, with an annual output of about 3500 tons and an annual market value of more than USD 2.5 billion. Exported Chinese royal jelly is sold in Japan, the United States and other regions [6,7].

RJ is a white, sticky, creamy substance that is secreted by the hypopharyngeal and mandibular glands of worker bees and is known as a “superfood”. Bee larvae eat RJ for the first three days after birth, and only queen bee larvae continue to eat RJ after three days. RJ helps the larvae to develop into a queen bee [5,8,9,10]. The pH of fresh RJ is usually between 3.6–4.2, water accounts for 60–70% (*w/w*) and it also contains protein, carbohydrates, vitamins, lipids, minerals, flavonoid compounds, polyphenols and other biologically active substances. The unique and rich active ingredients of RJ [11] contribute to its unique and rich pharmacological characteristics, such as antitumor, antiallergic, anti-inflammatory and immunomodulation effects [9]. Moreover, RJ does not cause serious side effects, such as diarrhea, in normal cases [12]. RJ was found to alleviate atopic dermatitis-like skin lesions in NC/Nga mice treated with picryl chloride by downregulating the production of TNP-specific IFN-γ and upregulating the expression of iNOS [13]. The effective uptake of antigens mediated by intestinally associated lymphoid tissue-specific M cells is a key step in inducing an effective intestinal mucosal immune response. Protease-treated RJ was reported to promote antigen-specific mucosal IgA response and regulate mucosal immune activity by enhancing the uptake of antigens by M cells in Caco-2 cells [14].

Intestinal diseases such as diarrhea are among the diseases with the highest morbidity and fatality rates in the world [15]. In 2016, diarrhea was the eighth leading cause of death in the world, with more than 1.6 million deaths globally. Diarrhea is also the fifth leading cause of death in children under five. In low-income and middle-income countries, diarrhea causes one in ten deaths among children under five years of age [16,17]. Inflammatory bowel disease (IBD) is a chronic inflammatory disease of the intestine that can cause abdominal pain and diarrhea and mainly characterized by the destruction of intestinal mucosal structure and changes in intestinal microbial composition. IBD mainly includes Crohn’s disease (CD) and ulcerative colitis (UC) [18,19]. Turan Karaca et al. [20,21,22] found that RJ alleviated different types of colitis. Treatment of acetic-acid-induced colitis with RJ alleviated the ulcerative erosion of colon tissue and increased colonic CD3^+^, CD45^+^ T cells and mast cells; RJ also increased the production of anti-inflammatory cytokine IL-10 and the activity of glutathione peroxidase (GSH-Px). RJ also reduced CD3^+^, CD5^+^, CD8^+^ and CD45^+^ T cells; the secretion of pro-inflammatory cytokines IL-1β and TNF-α; the key inflammatory mediators (COX-2 and NF-κB); and injury caused by tumor necrosis factor in rat colitis induced by 2,4,6-trinitrobenzene sulfonic acid (TNBS). To sum up, RJ has exhibited anti-inflammatory and antioxidant effects in artificially induced colitis [23]. The intestinal microbial flora of patients with IBD changes significantly [24]. As a functional food, RJ has been found to have antibacterial effects. Jelleines I-IV are important antibacterial components in RJ, and major royal jelly proteins 2–5 of RJ exhibit antibacterial activity against Gram-negative *E. coli* [3,25]. The effect of RJ on DSS-induced ulcerative colitis in mice has not yet been studied, so with our experiment, we intended to explore whether RJ has a protective effect on DSS-induced ulcerative colitis in mice.

## 2. Materials and Methods

### 2.1. Animal

A total of 50 5-week-old female C57BL/6JNifdc mice (14–17 g) were purchased from Vital River Laboratory Animal Technology Company (Beijing, China). All animals were raised in a controlled environment: relative humidity was 40 ± 10% and temperature was 20 ± 5 °C, with a 12/12 h light–dark cycle. After one week of adaptation, the mice were randomly divided into five groups, namely control group, DSS group and RJ treatment groups with three different contents, each with 10 mice. The RJ treatment groups treated with RJ were intragastrically administered 1.0 g/kg, 1.5 g/kg and 2.0 g/kg body weight RJ every day for 31 days (dissolved in normal saline, 10 mL/kg). The control group and DSS group were intragastrically administered normal saline (10 mL/kg) every day for 31 days. The DSS group and the RJ treatment group were given 3% w/v DSS drinking water on the 31st day for 7 consecutive days, and the control group was given drinking water without DSS. Body weight and disease activity index (DAI) scores of mice in each group were observed and recorded daily. The length of the colon was measured when the mice were sacrificed. All animal procedures were approved by the China Agricultural University Institutional Animal Care and Use Committee (AW32111202-2-3).

### 2.2. Royal Jelly Samples and Physicochemical Analysis

Samples of rape RJ were obtained from the Institute of Apicultural Research, Chinese Academy of Agricultural Sciences, and stored at −20 °C before use. The physicochemical parameters (water, protein, 10-hydroxy-2-decenoic acid, total sugar, starch, ash and acidity) of the RJ were measured according to the standard of General Administration of Quality Supervision, Inspection and Quarantine of the People’s Republic of China (GB9697-2008).

### 2.3. Disease Activity Index

After administration of DSS drinking water, the disease activity index score was recorded every day. The disease activity index score is a composite score of weight change, stool consistency and stool bleeding: weight loss percentage (0, 0; 1–%, 1; 5–15%, 2; >15%, 3), stool consistency (normal, 0; slightly loose, 1; loose, 2; diarrhea, 3), stool bleeding (normal, 0; slight, 1; modest, 2; severe, 3) [26].

### 2.4. Histologic Analysis and Periodic acid-Schiff Staining

After the mice were sacrificed, colon tissues were removed and fixed with 4% paraformaldehyde. Paraffin-embedded colon tissue sections were stained with hematoxylin and eosin (H.E.) for histopathological scoring. Histopathological score is a comprehensive measure of inflammatory cell infiltration and changes in tissue damage: inflammatory cell infiltration (rare inflammatory cells in the lamina propria, 0; increase in the number of inflammatory cells in the lamina propria, 1; inflammatory cells extending into the submucosa, 2; infiltration across the wall, 3) and tissue damage (no mucosal damage, 0; discrete lymphoepithelial damage, 1; surface mucosal erosion, 2; extensive mucosal damage and extension through deep structures of the intestinal wall) [27].

The changes in goblet cells in colon tissue were observed by periodic acid-Schiff staining. The number of goblet cells in each group is reflected by the mean of IOD.

### 2.5. FITC-Labeled Dextran Intestinal Permeability Assays

On the 7th day of DSS treatment, the mice were intragastrically administered 150 mg/mL, 60 mg/100 g body weight FITC-dextran after 5 h of fasting, and blood was taken from the eyeballs 4 h later. The blood was left to stand, then centrifuged at 3000 rpm/ for 10 min. The upper serum was diluted with PBS 10 times, and the fluorescence intensity was measured with a fluorescent spectrophotometer. The excitation light wavelength was 490 nm, and the emission light wavelength was 520 nm. By diluting FITC-dextran and setting different concentration gradients (0–4000 ng/mL), a standard curve was obtained.

### 2.6. Enzyme-Linked Immunosorbent Assay

Proteins were extracted from colon tissues with RIPA lysis buffer (CoWin Biotech Co., Inc., Beijing, China), then detected with a BCA protein analysis kit (CoWin Biotech Co., Inc., Beijing, China). The concentrations of secretory immunoglobulin (sIg)A, TNF-α (Laibotairui Tech Co., Ltd., Beijing, China), IL-10 and IL-6 (RayBiotech Life, Inc., Atlanta, GA, USA) were determined by an ELISA kit. The results were normalized to the protein concentration of each sample. A microplate reader (BioTek Co., Ltd., Beijing, China) was used for detection at 450 nm.

### 2.7. Western Blotting

Proteins extracted from the colon tissue were separated by 8–12% SDS-PAGE after the concentration was detected by the BCA protein assay kit (CW0014S, CoWin Biotech Co., Inc., Beijing, China). They were then transferred to 0.2 μm polyvinylidene fluoride membranes (Merck KGaA Co., Ltd., Darmstadt, Germany). Membranes were blocked with 5% skim milk for 1.5 h. After blocking, the membranes were incubated with anti-claudin 1 antibody (1/3000, Abcam Co., Inc., Cambridge, UK) and anti-claudin 3 antibody (1/1000, Abcam Co., Inc., Cambridge, UK) at 4 °C overnight. The membranes were washed with Tris-buffered saline Tween (TBST) and then incubated with horseradish peroxidase-conjugated goat anti-mouse IgG, or goat anti-rabbit IgG (CoWin Biotech Co., Inc., Beijing, China) for 1.5 h. The membranes were imaged with a Tanon 5200 imaging system (Tanon Science & Technology Co., Ltd., Shanghai, China).

### 2.8. Immunohistochemistry

Colon tissue sections were dewaxed with xylene and hydrated in gradient ethanol. Then, the antigen was repaired with 0.01 M sodium citrate buffer. Tissue sections were washed with phosphate buffer (PBS, pH7.0) 3 times for 5 min each time. Endogenous peroxidase activity was blocked by 3% hydrogen peroxide for 30 min, and non-specific staining was blocked by 5% goat serum for 30 min. Tissue sections were incubated with anti-occludin antibody (1/200) and anti-MUC2 (1/2000, Abcam Co., Inc., Cambridge, UK) at 4 °C overnight. On the second day, after being washed with PBS, they were incubated with biotin-conjugated goat anti-rabbit IgG (CoWin Biotech Co., Inc., Beijing, China) for 2 h and then incubated with horseradish peroxidase (HRP)-streptavidin (CoWin Biotech Co., Inc., Beijing, China) for 2 h. A DAB chromogenic reagent kit (Zhongshan Jinqiao Biotech Co., Ltd., Beijing, China) was used for chromogenic reagent detection. Hematoxylin was used for nuclear restaining. The primary antibody of the negative control group was replaced with PBS. Images were captured using a microscope (Nanjing Jiangnan Novel Optics Co., Ltd., Nanking, China).

### 2.9. TUNEL

Through the one step TUNEL apoptosis assay kit (Beyotime Biotech Co., Ltd., Shanghai, China), terminal deoxynucleotidyl transferase dUTP nick end labeling (TUNEL) assay was used to detect the apoptotic level of colon cells. The images were taken with upright DP72 microscope (Olympus Co., Inc., Tokyo, Japan).

### 2.10. Microbiota Analysis

The contents of the mouse colon were collected and stored in liquid nitrogen. An MN Nucleo Spin 96 Soi DNA extraction kit (MACHEREY-NAGEL GmbH & Co. KG, Duren, Germany) was used to extract total bacterial DNA from the sample. The primers were designed according to the conserved region of micro-organism V3+V4. The primers were used for PCR amplification, and the products were purified, quantified and homogenized to form a sequencing library. The constructed library was first subjected to quality inspection, and the qualified library was sequenced with Novaseq 6000 (Illumina, Co., Inc., San Diego, CA, USA). Sequence similarity greater than 97% was classified as an operational taxonomic unit (OTU). The OTU composition of different samples was analyzed using principal component analysis (PCA), principal coordinates analysis (PCoA) and non-metric multi-dimensional scaling (NMDS) based on Bray Curtis analysis. Line discriminant analysis (LDA) effect size, also known as LEfSe, was used to analyze the significance of differences between groups from the phylum to genus level. LEfSe analysis required an LDA score > 4.

### 2.11. Statistical Analyses

Data analysis was performed with GraphPad Prism (version 8.0.2 for Windows, GraphPad). Data are expressed as the means ± standard errors of the mean (SEMs). All comparisons of variant parameters between groups were made with one-way analysis of variance, with statistical significance as follows: * *p* < 0.05; ** *p* < 0.01; *** *p* < 0.001; the groups marked with different letters had significant differences.

## 3. Results

### 3.1. Analysis of Physicochemical Parameters of Royal Jelly

We first determined the main physicochemical parameters of RJ (water, protein, 10-hydroxy-2-decenoic acid, total sugar, acidity, etc.), and the results can be seen in Table 1. Water and 10-hydroxy-2-decenoic acid (10-HDA) contents are important parameters for evaluating the quality of RJ. The International Organization for Standardization (ISO) stipulates that the minimum content of 10-HDA in RJ is 1.4% (ISO12824:2016), and the General Administration of Quality Supervision, Inspection and Quarantine of the People’s Republic of China stipulates that the 10-hydroxy-2-decenoic acid(10-HDA) content in qualified RJ products should be higher than 1.4% (GB9697-2008). The water content required by ISO (ISO12824:2016) is 62.0–68.5%. In China, the water content requirements of qualified RJ products and high-quality RJ products are <69.0% and <67.5%, respectively, and there is no lower limit (GB9697-2008). China also requires a protein content of 11–16%, total sugar ≤ 15%, ash content ≤ 1.5% and acidity of 30–53 mL/100 g, and starch must not be detected. Our test results showed that 10-HDA and moisture content were in line with international and Chinese quality standards; other components met Chinese quality standards, and the moisture content even met the Chinese standards for high-quality RJ products [28].

### 3.2. Royal Jelly Alleviated Acute Colitis Induced by DSS in Mice

The experimental treatment is shown in Figure 1A. After three days of DSS drinking water treatment, the weight of the mice gradually decreased (*p* < 0.01) (Figure 1B), the length of the colon was shortened (*p* < 0.0001) (Figure 1D,E) and the disease activity index score gradually increased (*p* < 0.001) (Figure 1C). On the seventh day of DSS treatment, oral administration of 2.0 g/kg RJ significantly improved the weight change of mice (*p* < 0.05) (Figure 1B) but also improved the disease activity index score (*p* < 0.05) (Figure 1C) and the change of colon length (*p* = 0.0012) (Figure 1D,E). RJ at other concentrations also had a certain alleviating effect. These results indicate that the establishment of a mouse model of ulcerative colitis was successful and that RJ alleviated the symptoms of ulcerative colitis caused by DSS.

### 3.3. Royal Jelly Reduced Colon Damage in Acute Colitis Induced by DSS in Mice

After DSS treatment, the mouse colon tissue showed inflammatory cell infiltration and mucosal damage, and the histopathological score was significantly increased (*p* < 0.0001) (Figure 2A,B). Different concentrations of RJ improved the colon damage caused by DSS to varying degrees, and 2.0 g/kg RJ significantly reduced the histopathological score (*p* = 0.0171) (Figure 2A,B). TUNEL staining results showed that DSS caused an increase in colonic epithelial cell apoptosis, whereas 2.0 g/kg RJ reduced cell apoptosis (Figure 2D). Apoptotic cells are labeled with green fluorescence. The apoptosis of intestinal epithelial cells mainly occurred at the top of the mucosal epithelium and intestinal crypt. In conclusion, royal jelly alleviated colonic injury induced by DSS in mice with acute colitis, mainly by reducing the integrity of colonic tissue structure.

### 3.4. Royal Jelly Improved Intestinal Permeability and Inflammation in Mice with DSS-Induced Acute Colitis

The DSS treatment significantly increased intestinal permeability (*p* = 0.0006), whereas RJ significantly decreased intestinal permeability (*p* < 0.05) (Figure 2C). Tight-junction proteins claudin 1, claudin 3 and occludin (*p* = 0.0003), which are closely related to intestinal permeability, decreased. These proteins increased following treatment with 2.0 g/kg RJ (claudin1: *p* = 0.0538; occludin: *p* = 0.0161) (Figure 3). In addition, the mucin layer on the epithelial surface of the intestinal mucosa is also an important part of the intestinal mechanical barrier. MUC2, a mucin in the mucin layer, and its secreting goblet cells were also significantly reduced. The use of RJ significantly increased the expression of these proteins. Among them, the concentration of 2.0 g/kg RJ played a significant role (Figure 4) (MUC2: *p* = 0.0575; GC: *p* = 0.0009). Changes in the expression of these important proteins after oral administration of RJ suggested that RJ reduced colon injury by protecting intestinal permeability. We further analyzed the expression levels of inflammatory cytokines in colon tissues. The results showed that the DSS treatment caused an increase in the expression of the colonic proinflammatory cytokines IL-6 and TNF-α, as well as a decrease in the expression of anti-inflammatory cytokines IL-10 and sIgA (no significant decrease in sIgA was observed), whereas RJ improved the expression of these cytokines (IL-10: *p* = 0.9994; IL-6: *p* = 0.0361; TNF-α: *p* = 0.0120; sIgA: *p* = 0.0190) (Figure 5). This also suggested that RJ might reduce colon damage by reducing intestinal inflammation.

### 3.5. Gut Microbiota Profiling

The 16SrRNA gene sequencing method was used to detect the colonic flora in each treatment group, and the sequence similarity in OTU was greater than 97%. The Veen diagram of OTUs showed that the colonic flora of each group of mice contained 350 common OTUs, and except for the control group, which had five unique OTUs; the other groups had no unique OTUs (Figure 6A). Principal component analysis (PCA), principal coordinates analysis (PCoA) and non-metric multi-dimensional scaling (NMDS) based on Bray–Curtis analysis can reflect the differences and distances of samples by analyzing the composition of OTUs (97% similarity) of different samples. The closer the distance between two samples, the more similar the composition of the samples. The results of PCA (Figure 6B), PCoA (Figure 6C) and NMDS (Figure 6D) showed that compared with the DSS group, the 2.0 g/kg RJ treatment group and the control group had more similar intestinal flora compositions, and the distance was closer. The abundance comparison chart of the significant analysis of differences between groups showed that DSS caused *Parabacteroides* (Figure 6E), *Erysipelotrichaceae* (Figure 6F) and *Proteobacteria* (Figure 6G–J) (*Gammaproteobacteria*, *Enterobacteriales*, *Enterobacteriaceae*), and the relative abundance of harmful bacteria *Escherichia Shigella* (Figure 6K) of colon increased. The relative abundance of beneficial bacteria *Muribaculum* (Figure 6L) also decreased. In the RJ treatment group, the relative abundance of the above intestinal flora was improved following treatment with 2.0 g/kg RJ (Figure 6E-L). In conclusion, the use of royal jelly changed the composition of intestinal flora.

## 4. Discussion

As a functional product, RJ is rich in proteins, carbohydrates, vitamins, lipids, minerals, flavonoids, polyphenols and other biologically active substances [11]. Our analysis of physicochemical composition indicated that the rape RJ used in this experiment contained 62.4% water, 14.2% protein, 11.1% total sugars and 1.51% 10-HDA. Its acidity was 33.5 mL/100 g. This was consistent with previous research results and in compliance with international and Chinese standards [28,29]. 10-HDA is the main and unique lipid component in RJ and can be used as an important indicator to verify the freshness of RJ [30,31]. Active ingredients, such as 10-HDA and major royal jelly proteins, play an important role in anticancer, antibacterial, immune regulation, etc. [32]. IBD includes CD and UC [33]. The DSS-induced mouse colitis model has been widely used in the study of ulcerative colitis and IBD [34]. In this study, DSS treatment caused weight loss, colon shortening and an increase the disease activity index score in mice. These results indicated that the mouse model of ulcerative colitis was successfully established, and RJ improved the symptoms of ulcerative colitis caused by DSS.

The intestinal mucosal epithelial barrier includes intestinal mucosal mechanical barrier, the intestinal mucosal immune barrier and the intestinal mucosal microbial barrier. The intestinal mechanical barrier is composed of intestinal mucosal epithelial cells, tight junctions between epithelial cells and the mucous layer on the surface of the mucosal epithelium. It is the key tissue structure of the intestine involved in resistance to invasion of bacteria and pathogens, maintaining the intestinal epithelial barrier function and select permeability. The mucus layer protects against invading pathogens and microorganisms by secreting mucin from goblet cells [35]. There is an interaction between mucin and micro-organisms. The change of microorganisms with mucus-degrading activity regulates the diversity of intestinal microbes and causes changes in the permeability of the mucus layer and mucus. The permeability of the mucous layer increased when the mineralization of *Desulfovibrio*, *Bacteroides*, *Parabacteroides* and *Prevotella* occurred with mucous degradation ability [36]. DSS induced apoptosis of colonic epithelial cells, whereas RJ reduced the number of apoptotic cells. The DSS treatment resulted in a significant decrease in the number of goblet cells, followed by a significant decrease in the expression of mucin, and RJ significantly increased their expression. Tight-junction proteins, including occludin, junction-adhesion molecules and members of the claudin family can connect adjacent epithelial cells. Occludin and claudins interact with the zonula occludens proteins attached to the actin cytoskeleton and, together, play a crucial role in paracellular permeability. The defect of the intestinal mucosal barrier allows the intestinal cavity antigens to enter the lamina propria, thereby increasing the susceptibility to IBD [37]. In DSS-induced colitis in mice, intestinal permeability increased significantly, and tight-junction proteins occludin, claudin 1 and claudin 3 that are closely related to intestinal permeability decreased, whereas RJ significantly improved the changes of intestinal permeability. Plasma cells in the lamina propria can secrete secretory IgA (sIgA) into the intestinal lumen. sIgA is the most abundant and representative immunoglobulin on the mucosal surface, which can prevent microbial antigens from entering the intestinal epithelial cells [35,38]. DSS caused a decrease in the expression of sIgA and anti-inflammatory factors and an increase in the expression of proinflammatory factors, whereas RJ had the opposite effect.

*Bacteroidetes*, *Firmicutes*, *Actinobacteria* and *Proteobacteria* account for nearly 99% of the gut microbiota of healthy individuals, and about 90% of the gut microbiota are *Firmicutes* and *Bacteroidetes* [39]. *Firmicutes* and *Bacteroidetes* produce short-chain fatty acids (SCFA) by digesting dietary plant fibers that humans cannot. Short-chain fatty acids, especially butyrate, are the main energy source for colon epithelial cells [40]. The intestinal flora of patients with IBD are mainly realized as a decrease in biodiversity and an increase in *Proteobacteria* (such as *Enterobacteriaceae* and *Bilophila*) and certain *Bacteroidetes* [41]. The experimental results show that *Proteobacteria* (*Gammaproteobacteria*, *Enterobacteriales* and *Enterobacteriaceae*) and certain *Bacteroidaceae* (*Parabacteroides*) in the colon of DSS-induced colitis mice did increase. The increase in the abundance of *Parabacteroides* in the colon might be one of the reasons for the decrease in goblet cells and mucin MUC2 in the colon mucous layer of mice with DSS-induced colitis, as well as the increase in intestinal permeability. RJ might improve colonic permeability by decreasing the abundance of *Parabacteroides*, thus improving mucosal barrier damage. After DSS treatment, the relative abundance of *Erysipelotrichaceae* in the colon increased, whereas RJ reduced its relative abundance. Our experimental results are consistent with the results of previous studies, which also found that *Erysipelotrichaceae* of *Firmicutes* were significantly increased during the acute inflammatory period [42]. The relative abundance of harmful bacteria *Escherichia Shigella* and beneficial bacteria *Muribaculum* in the colon of mice with DSS-induced colitis was also regulated by RJ.

## 5. Conclusions

In summary, as a functional food, RJ could alleviate acute colitis induced by DSS by improving the colonic mucosal barrier and gut microbiota. Further studies are needed to confirm the specific mechanism of the relieving effect of royal jelly on colitis mice, as well as the active ingredients that played a role in this process.

## Figures and Tables

**Figure 1 nutrients-14-02069-f001:**
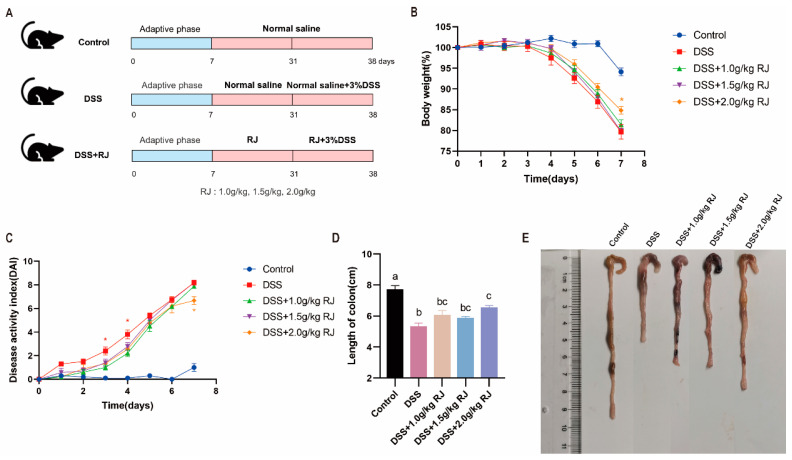
Oral administration of royal jelly improved the symptoms of DSS-induced ulcerative colitis in mice. (**A**) Experimental treatment and grouping; (**B**) weight loss (*n* = 8 for DSS + 2.0 g/kg RJ group, *n* = 10 for other groups); (**C**) disease activity index score changes; (**D**,**E**) colon length (*n* = 8). A dose of 3% *w/v* DSS drinking water was given on day 31, which was recorded as day 0. Data were expressed as mean ± standard error (SEM). “ * ” means *p* < 0.05, and there is significant difference between the two groups. The groups marked with different letters had significant differences. The red asterisk (*) indicates that the DSS group was significantly different from the RJ treatment group, and the yellow asterisk (*) indicates that the DSS + 2.0 g/kg RJ group was significantly different from the DSS group.

**Figure 2 nutrients-14-02069-f002:**
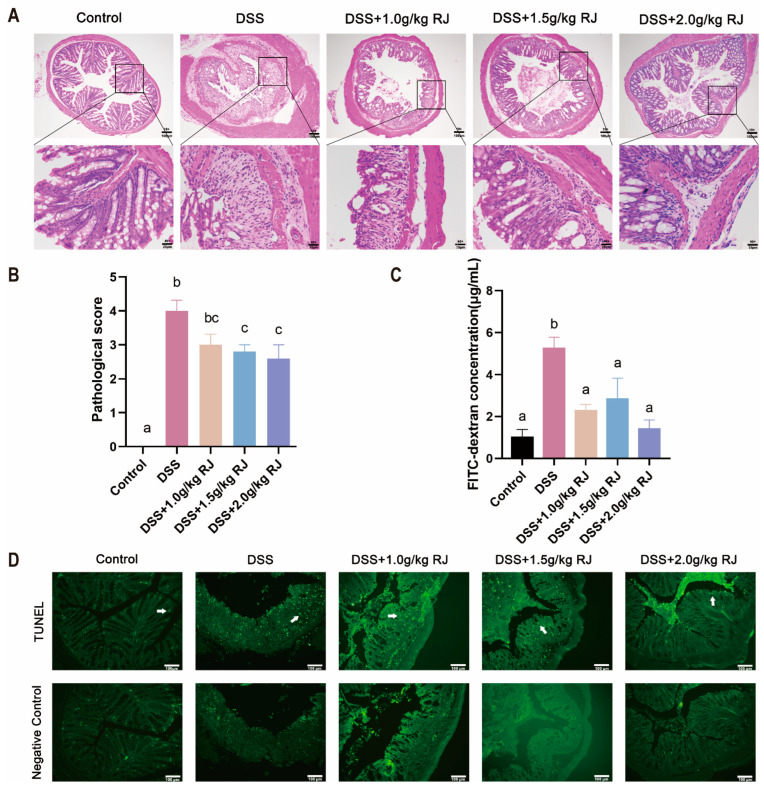
Oral administration of royal jelly improved the colon damage induced by DSS in mice with ulcerative colitis. (**A**) HE staining of colon tissue. The first line of the scale bar is 100 µm, and the second line of the scale bar is 25 µm. (**B**) Histopathological score of colon (*n* = 5). (**C**) Changes in intestinal permeability (*n* = 4). (**D**) TUNEL staining (*n* = 3). The white arrow points to apoptotic cells. The scale bar is 100µm. Data are expressed as mean ± standard error (SEM). The groups marked with different letters had significant differences.

**Figure 3 nutrients-14-02069-f003:**
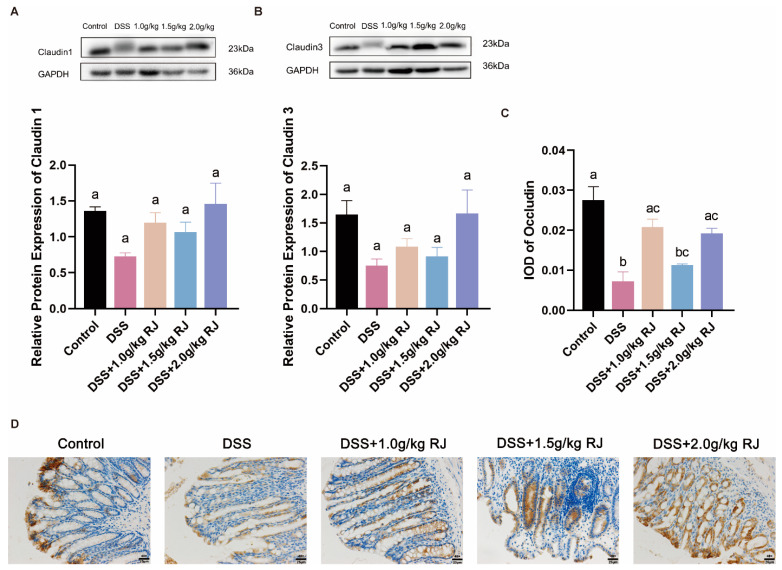
Oral administration of royal jelly increased the expression of tight-junction protein in the colonic mucosa of mice with ulcerative colitis induced by DSS. Protein bands and statistical analysis of colonic tight-junction proteins claudin 1 (**A**) and claudin 3 (**B**) by Western blotting (*n* = 3); Immunohistochemical staining (**C**) and mean of IOD statistics (**D**) of colonic tight-junction protein occludin (*n* = 3). The scale bar is 25 µm. Data were expressed as mean ± standard error (SEM). The groups marked with different letters had significant differences.

**Figure 4 nutrients-14-02069-f004:**
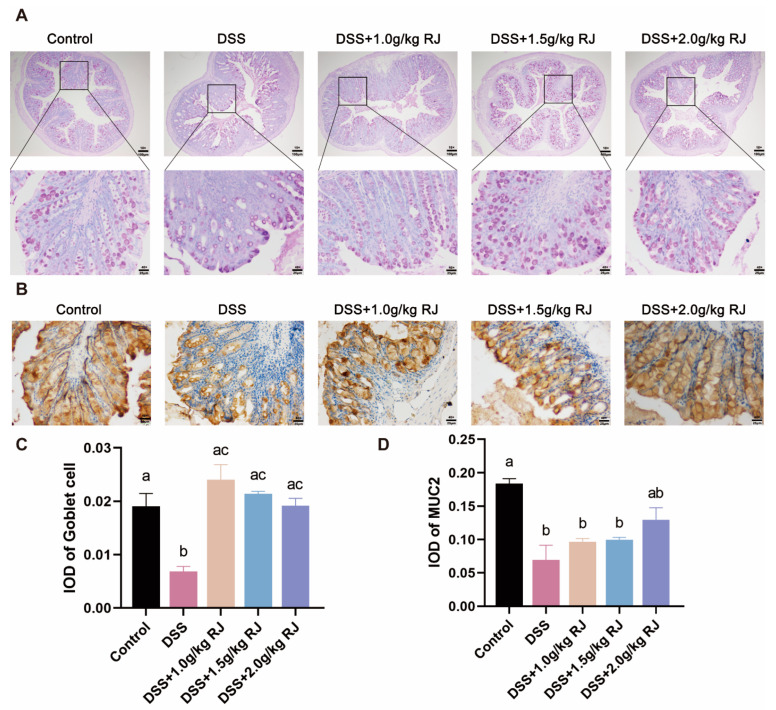
Oral administration of royal jelly increased the expression of goblet cells and their secretion mucin, MUC2, in the colon mucosa of mice with ulcerative colitis induced by DSS. (**A**) PAS staining of colonic goblet cells (the first line of the scale bar is 100µm, and the second line of the scale bar is 25 µm). (**B**) Mucin MUC2 was stained by immunohistochemistry (the scale bar is 25 µm). IOD was counted of goblet cell by PAS staining (**C**) (*n* = 3 for DSS + 1.0 g/kg RJ group, *n* = 4 for other groups) and the MUC2 by immunohistochemical staining (**D**) (*n* = 3). Data are expressed as mean ± standard error (SEM). The groups marked with different letters had significant differences.

**Figure 5 nutrients-14-02069-f005:**
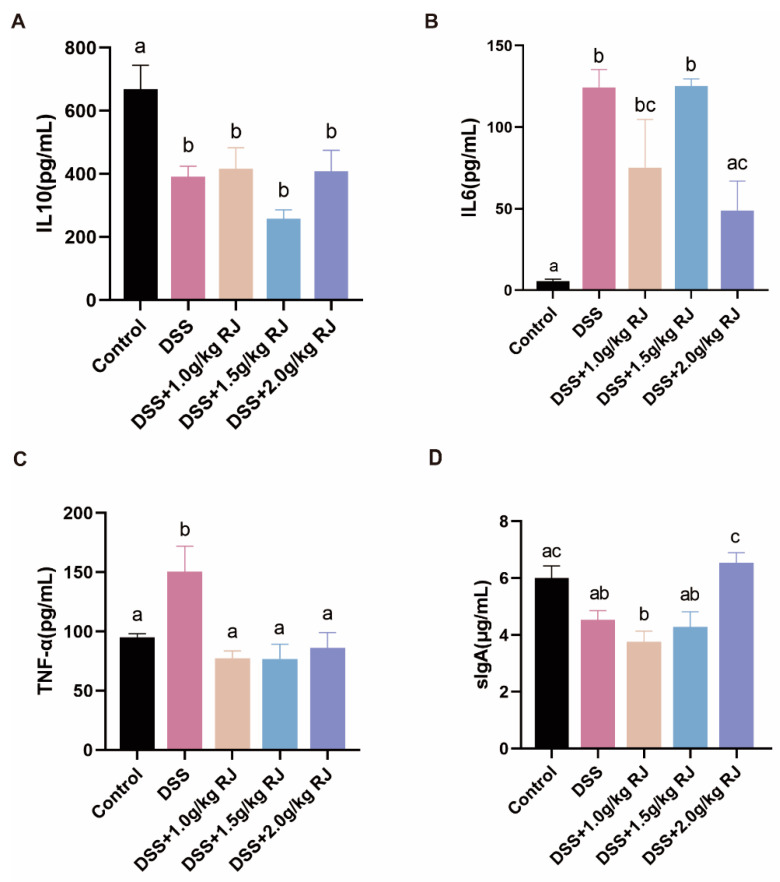
Oral administration of royal jelly reduced the content of proinflammatory cytokines in DSS-treated mice and increased the content of anti-inflammatory cytokines and sIgA. (**A**–**D**) Concentrations of IL-10 (*n* = 5), IL-6 (*n* = 3–4), TNF-α (*n* = 5–6) and sIgA (*n* = 5) in colon tissue, respectively. Data are expressed as mean ± standard error (SEM). The groups marked with different letters had significant differences among groups.

**Figure 6 nutrients-14-02069-f006:**
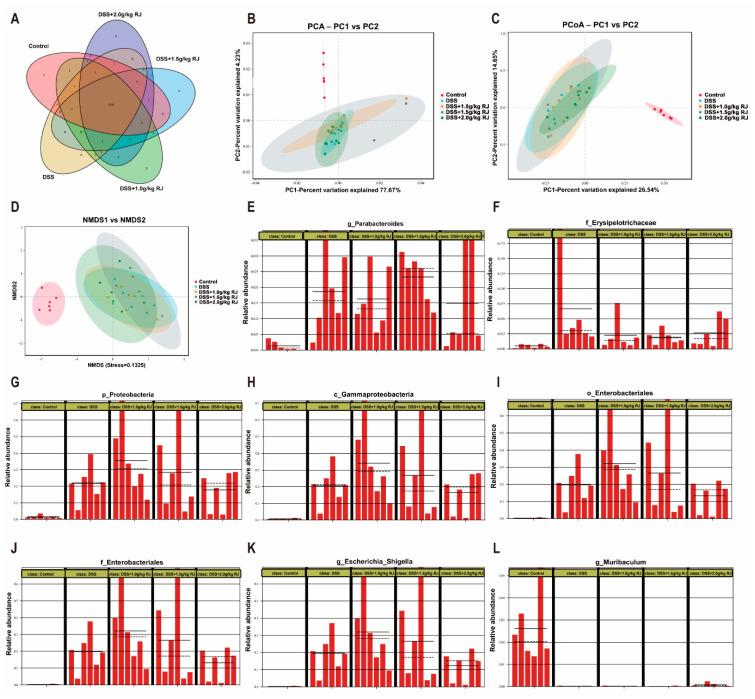
Oral administration of royal jelly altered the microbial composition of the colon in DSS-treated mice. (**A**) Veen diagram of colonic microorganisms OTUs. (**B**) Principal component analysis (PCA) of colonic microorganisms. (**C**) Principal coordinates analysis (PCoA) of colonic microorganisms. (**D**) Non-metric multi-dimensional scaling (NMDS) of colonic microorganisms; comparison diagram of abundance of *Parabacteroides* (**E**), *Erysipelotrichaceae* (**F**) and *Proteobacteria* (**G**–**J**) (*Gammaproteobacteria*, *Enterobacteriales* and *Enterobacteriaceae*), *Escherichia Shigella* (**K**) and *Muribaculum* (**L**) for significance analysis of differences between groups. The solid and dashed lines represent the average and median of the relative abundance in each group of samples (*n* = 6).

**Table 1 nutrients-14-02069-t001:** Physicochemical parameters of royal jelly.

Variable	Unit	Royal Jelly
Water	g/100 g	62.4
Protein	g/100 g	14.2
10-HDA	g/100 g	1.51
Total sugar	g/100 g	11.1
Amylum	g/100 g	Not detected
Ash	g/100 g	0.4
Acidity	mL/100 g	33.5

## Data Availability

The data presented in this study are available on request from the corresponding author.

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
