# Peer review of "Royal Jelly Protected against Dextran-Sulfate-Sodium-Induced Colitis by Improving the Colonic Mucosal Barrier and Gut Microbiota"

_nutrients, 2022, doi:10.3390/nu14102069_

Round 1
Reviewer 1 Report
Overall comments: The work by Guo et al. is a very significant work. There are some inappropriate citations in the manuscript that are not relevant. I have mentioned one of them below. I have detected some plagiarism issues. Authors should change or alter those sentences. I have detected some minor English mistakes. Scientifically the article is well organized and written. The introduction is well written. Materials and methods are clear. In the figures, scale bars should be given. The manuscript can be considered for publication only after careful revision.
Specific comments:
Line 25: Escherichia Shigella of colon to increase…should be…Escherichia Shigella in colon to increase
Line 33-35: More and more natural products are used in health problems at present, which have been proved to reduce the risk of patients with chronic diseases and the investment in health care…I think the wrong reference has been cited here. Please replace with https://doi.org/10.1038/s41573-020-00114-z
Line 35-37: In the past few years, natural bee products have been widely applied in traditional and modern medicine. Royal jelly is marketed on a large scale in Asia, especially in China and Japan, as a health food and cosmetics [2]…A very recent paper could be added here…https://doi.org/10.1021/acs.jafc.1c05822
Line 91: 1.0g/kg, 1.5g/kg and 2.0g/kg…there is some punctuation missing. Please check it throughout the manuscript.
Line 157: for the units, please follow the journal instruction. For example, minutes…According to the author's guideline, should it have been min?
Figures 2, 3, 4: scale bars should be given. It’s easy to understand the readers. Figure 3 D, the figures are not clear. Please upload the high-resolution photos. Figure 6 D, the writing is not readable.
Line 366: I have detected some plagiarism, for example, In most healthy individuals, 99% of the gut microbiota consists of Firmicutes, Bacteroidetes, Proteobacteria, and Actinobacteria, of which Firmicutes and Bacteroidetes together account for approximately 90% of the microbiota
Author Response
Detailed Response to Reviewer
Dear reviewer of Nutrients:
We are very grateful to you for your valuable opinions. We have revised our manuscript “Royal Jelly protected against dextran sulfate sodium-induced colitis via improving the colonic mucosal barrier and gut microbiota” and entitled it “nutrients-1654418 R1”. We would like to thank reviewers for your appreciation of the article. We have studied comments carefully and have made correction which we hope meet with approval. All revised portions are marked in red in the manuscript (nutrients-1654418 R1). The main corrections in the paper and the responds point to your comments are as following.
Finally, thanks again to your suggestions.
Corresponding authors:
Name: Yulan Dong
E-mail: [email protected]
Co-Corresponding authors:
Name: Wenli Tian
E-mail: [email protected]
We look forward to your reply. Thank you very much for consideration and with best regards!
Sincerely yours
Yulan Dong
Reviewer #1:
Question 1: Line 25: Escherichia Shigella of colon to increase…should be…Escherichia Shigella in colon to increase.
Answer: Thank you very much for pointing out the mistake. We have corrected it.
Please see line 25 of page 1 in the manuscript R1.
Question 2: Line 33-35: More and more natural products are used in health problems at present, which have been proved to reduce the risk of patients with chronic diseases and the investment in health care…I think the wrong reference has been cited here. Please replace with https://doi.org/10.1038/s41573-020-00114-z
Answer: Thank you very much for your advice. We have carefully checked the manuscript and the reference quoted here, and the description about natural products are indeed described in the reference quoted. This reference points out that natural products are increasingly used worldwide for health benefits, and have demonstrated physiological benefits, reduced risk of chronic diseases and provided savings in healthcare. At the same time, the reference you mentioned also described natural products very well, so I added this reference on the basis of retaining the original reference. Thank you very much for your comment again.
Please see line 35 of page 1 and line 413-414 of page 13 in the manuscript R1.
Question 3: Line 35-37: In the past few years, natural bee products have been widely applied in traditional and modern medicine. Royal jelly is marketed on a large scale in Asia, especially in China and Japan, as a health food and cosmetics [2]…A very recent paper could be added here…https://doi.org/10.1021/acs.jafc.1c05822
Answer: Thank you very much for your advice. We have added it to the manuscript.
Please see line 36-37 of page 1 and line 417-419 of page 13 in the manuscript R1.
Question 4: Line 91: 1.0g/kg, 1.5g/kg and 2.0g/kg…there is some punctuation missing. Please check it throughout the manuscript.
Answer: Thank you very much for pointing out the mistake. We have also corrected it. And We checked the full text of the manuscript. We found some minor errors and corrected.
Please see line 93 of page 2, line 138 of page 3, line 148 of page 4 and line 399 of page 13 in the manuscript R1.
Modification of space between number and symbol, please see line 27 of page 1, line 206, 215- 220 of page 5, line 226-227, 231, 235, 237-239, 246-250, 254-255 of page 6, line 261-262, 267-268 of page 7, line 274, 276 of page 8, line 283-284 of page 9, line 289, 303 of page 10, line 311, 322 of page 11 in the manuscript R1.
Question 5: Line 157: for the units, please follow the journal instruction. For example, minutes…According to the author's guideline, should it have been min?
Answer: Thank you very much for pointing out the mistake. We read the journal instructions and changed minutes to min and hour to h according to their requirements.
Please see line 128-129 of page 3, line 158-160, 164 of page 4 in the manuscript R1.
Question 6: Figures 2, 3, 4: scale bars should be given. It’s easy to understand the readers. Figure 3D, the figures are not clear. Please upload the high-resolution photos. Figure 6D, the writing is not readable.
Answer: Thank you very much for your advice. The scale in Figures 2, 3, and 4 may not be very clear, so we modified and bolded the scale. In the case of Figure 3D and Figure 6D, we changed them to a clearer picture.
Please see Figure 2 line 264 of page 7, Figure 3 line 271 of page 8, Figure 6 line 313 of page 11 in the manuscript R1.
Question 7:I have detected some plagiarism, for example, In most healthy individuals, 99% of the gut microbiota consists of Firmicutes, Bacteroidetes, Proteobacteria, and Actinobacteria, of which Firmicutes and Bacteroidetes together account for approximately 90% of the microbiota
Answer: Thanks a lot for pointing out the error. We have revised the description here, similar to the reference.
Please see line 367-369 of page 12 in the manuscript R1.
Reviewer 2 Report
Manuscript “nutrients-165441” describes the protected effect of royal jelly against DSS-induced colitis. The findings are very interesting
However, I would like a comment by the authors regarding the quantity of RJ required to achieve such results for two distinct reasons. Firstly, 2g / Kg is excessively high for human consumption, as recommended dosages can be up to 1 g per day. Secondly, since RJ is a high-priced product, 2g / Kg is economically prohibitive to be used by humans.
Please, feel free either to include your comments to the manuscript or just to answer through your reply to reviewers’ comments.
Minor comments
Lines 46-48. Rewrite as follows: Regarding fresh RJ, pH is usually between 3.6-4.2, water accounts for 60-70% (w/w), while also it contains protein, carbohydrate, vitamins, lipids, minerals, flavonoid compounds, polyphenols and other biologically active substances.
Line 67. Delete “etc” since you used the word “mainly” earlier.
Line 90. Please, make clear that the administration of RJ was daily.
Section 2.2. and 3.1. In section 2.2. it is implied that you used more than one RJ sample, yet in section 3.1. results are given for one. Please, correct this inconsistency. If you used one bulk sample of RJ, 3.1. is OK, but 2.2. needs to be corrected. If you used more than one samples (more probable), give min and max levels, as well SD in Table 1.
Line 324. The fact that RJ was of rape origin should be indicated in section 2.2.
Line 329. Delete “acid”.
Line 330-331. Something is wrong with this sentence.
Reference section. Correct Journal abbreviations when the word is not fully given. Either use a full stop (.) or not. Even though it is not clear in the Journal’s “Instruction for Authors”, it seems that a full stop should be used.
Author Response
Detailed Response to Reviewer
Dear reviewer of Nutrients:
We are very grateful to you for your valuable opinions. We have revised our manuscript “Royal Jelly protected against dextran sulfate sodium-induced colitis via improving the colonic mucosal barrier and gut microbiota” and entitled it “nutrients-1654418 R1”. We would like to thank reviewers for your appreciation of the article. We have studied comments carefully and have made correction which we hope meet with approval. All revised portions are marked in red in the manuscript (nutrients-1654418 R1). The main corrections in the paper and the responds point to your comments are as following.
Finally, thanks again to your suggestions.
Corresponding authors:
Name: Yulan Dong
E-mail: [email protected]
Co-Corresponding authors:
Name: Wenli Tian
E-mail: [email protected]
We look forward to your reply. Thank you very much for consideration and with best regards!
Sincerely yours
Yulan Dong
Reviewer #2:
Question 1: However, I would like a comment by the authors regarding the quantity of RJ required to achieve such results for two distinct reasons. Firstly, 2 g/kg is excessively high for human consumption, as recommended dosages can be up to 1 g per day. Secondly, since RJ is a high-priced product, 2 g/kg is economically prohibitive to be used by humans.
Please, feel free either to include your comments to the manuscript or just to answer through your reply to reviewers’ comments.
Answer: In mice, 1 g/kg and 1.5 g/kg of royal jelly were widely used in some references. According to study of Reagan-Shaw et al (https://doi.org/10.1096/fj.07-9574LSF), the consumption of 1g /kg royal jelly in mice corresponds to the recommended daily consumption of 3-5g for healthy human. In this paper, royal jelly is used as adjuvant therapy, and its consumption can be appropriately increased. 2 g/kg royal jelly used in this paper is mainly used for DSS induced acute ulcerative colitis model. As a short-term dosage, it is very important for the prevention of disease in healthy people and patients with inflammatory bowel disease after recovery (the recurrence rate of inflammatory bowel disease is high). The price of royal jelly on the market in China is about 250 yuan (RMB)/500g. According to 5g/day consumption of human, the daily consumption is about 2.5 yuan (RMB). 2g/kg of mice corresponds to about 10g/ day for human, and the daily consumption is about 5 yuan (RMB), which is a small burden for people's life and can be accepted. Ahmad et al (https://doi.org/10.3390/ijms21020382) found that royal jelly is sold in large quantities in China and other Asian countries.
Question 2: Lines 46-48. Rewrite as follows: Regarding fresh RJ, pH is usually between 3.6-4.2, water accounts for 60-70% (w/w), while also it contains protein, carbohydrate, vitamins, lipids, minerals, flavonoid compounds, polyphenols and other biologically active substances.
Answer: We deeply appreciated your comment of our manuscript. We have also corrected it.
Please see line 48 - 51 of page 2 in the manuscript R1.
Question 3: Line 67. Delete “etc” since you used the word “mainly” earlier.
Answer: Thank you very much for pointing out the mistake. We have corrected it.
Please see line 68 of page 2 in the manuscript R1.
Question 4: Line 90. Please, make clear that the administration of RJ was daily.
Answer: Thank you very much for pointing out the mistake. We have also corrected it
Please see line 92 to 96 of page 2 in the manuscript R1.
Question 5: Section 2.2. and 3.1. In section 2.2. it is implied that you used more than one RJ sample, yet in section 3.1. results are given for one. Please, correct this inconsistency. If you used one bulk sample of RJ, 3.1. is OK, but 2.2. needs to be corrected. If you used more than one samples (more probable), give min and max levels, as well SD in Table 1.
Answer: Thank you very much for your advice. We used one bulk sample of RJ, so we corrected Section 2.2.
Please see line 103 of page 3 in the manuscript R1.
Question 6: Line 324. The fact that RJ was of rape origin should be indicated in section 2.2.
Answer: We deeply appreciate your comment of our manuscript. We have added the origin of RJ in section 2.2.
Please see line 103 of page 3 in the manuscript R1.
Question 7: Line 329. Delete “acid”.
Answer: We deeply appreciated your comment of our manuscript. We accepted your suggestion and revised it.
Please see line 331 of page 11 in the manuscript R1.
Question 8: Line 330-331. Something is wrong with this sentence.
Answer: Thank you very much for pointing out the mistake. We have also corrected it.
Please see line 332 of page 12 in the manuscript R1.
Question 9: Reference section. Correct Journal abbreviations when the word is not fully given. Either use a full stop (.) or not. Even though it is not clear in the Journal’s “Instruction for Authors”, it seems that a full stop should be used.
Answer: Thank you very much for pointing out the mistake. We read other articles in the journal and modified them according to their format.
Please see line 416, 423, 425 of page 13 and line 438, 459 of page 14 in the manuscript R1.